# Collateral Sensitivity to Fosfomycin of Tobramycin-Resistant Mutants of *Pseudomonas aeruginosa* Is Contingent on Bacterial Genomic Background

**DOI:** 10.3390/ijms24086892

**Published:** 2023-04-07

**Authors:** Roberta Genova, Pablo Laborda, Trinidad Cuesta, José Luis Martínez, Fernando Sanz-García

**Affiliations:** 1Centro Nacional de Biotecnología, CSIC, 28043 Madrid, Spain; 2Department of Biotechnology and Environmental Protection, Estación Experimental del Zaidín, CSIC, 18008 Granada, Spain; 3The Novo Nordisk Foundation Center for Biosustainability, Technical University of Denmark, 2800 Kongens Lyngby, Denmark; 4Department of Clinical Microbiology 9301, Rigshospitalet, 2100 Copenhagen, Denmark; 5Microbiology Department, Medicina Preventiva y Salud Pública, Universidad de Zaragoza, 50009 Zaragoza, Spain

**Keywords:** collateral sensitivity, fosfomycin, *Pseudomonas aeruginosa*, tobramycin, evolution contingency, antibiotic resistance, bacterial evolution

## Abstract

Understanding the consequences in bacterial physiology of the acquisition of drug resistance is needed to identify and exploit the weaknesses derived from it. One of them is collateral sensitivity, a potentially exploitable phenotype that, unfortunately, is not always conserved among different isolates. The identification of robust, conserved collateral sensitivity patterns is then relevant for the translation of this knowledge into clinical practice. We have previously identified a robust fosfomycin collateral sensitivity pattern of *Pseudomonas aeruginosa* that emerged in different tobramycin-resistant clones. To go one step further, here, we studied if the acquisition of resistance to tobramycin is associated with robust collateral sensitivity to fosfomycin among *P. aeruginosa* isolates. To that aim, we analyzed, using adaptive laboratory evolution approaches, 23 different clinical isolates of *P. aeruginosa* presenting diverse mutational resistomes. Nine of them showed collateral sensitivity to fosfomycin, indicating that this phenotype is contingent on the genetic background. Interestingly, collateral sensitivity to fosfomycin was linked to a larger increase in tobramycin minimal inhibitory concentration. Further, we unveiled that *fosA* low expression, rendering a higher intracellular accumulation of fosfomycin, and a reduction in the expression of the *P. aeruginosa* alternative peptidoglycan-recycling pathway enzymes, might be on the basis of the collateral sensitivity phenotype.

## 1. Introduction

The antibiotic resistance of bacterial pathogens is currently one of the most relevant threats to human health [1,2], a problem that has historically been addressed by a continuous development of novel antibiotics. However, the discovery of new categories of structurally different antimicrobials has been alarmingly scarce in the recent past. Hence, in this situation, a better use of the current repertoire of antibiotics is mandatory to curb the problem of resistance. For this purpose, recent works on the matter have proposed that the exploration of the evolutionary trade-offs associated with the acquisition of drug resistance could be a feasible strategy to implement knowledge-based approaches to tackle antibiotic resistance [3,4,5,6]. One of these potentially exploitable trade-offs is collateral sensitivity. This phenotype, defined several decades ago, is a phenomenon by which the acquisition of resistance to one drug leads to an increased susceptibility to another one [7]. This information is useful to define drug pairs that may be used, either sequentially or in combination, thereby improving their therapeutic potential [8]. Numerous works have explored the collateral sensitivity networks in bacterial pathogens [3,9,10,11,12]. However, the main problem that arises when it comes to their application is the robustness of this hypersusceptibility phenotype in bacteria presenting different genomic backgrounds, which exemplify the heterogeneous populations that are usually responsible of chronic infections in humans [13]. Indeed, while some—few—robust susceptibility patterns have been identified, on several occasions, the phenotype is not conserved even among replicated cultures from the same bacterial clones [14,15,16]. This indicates that collateral sensitivity may hold a certain degree of stochasticity and might be contingent on the genomic background. Taking this into account, it is important to establish which collateral sensitivity patterns are robust—hence useful for defining better therapeutic strategies—and which ones are not, in clinically relevant bacterial pathogens.

Among the pathogens whose antibiotic resistance entails utmost concern, *Pseudomonas aeruginosa* stands out. This bacterium is one of the most prevalent nosocomial pathogens, given its role as a recurring artificer of chronic infections in cystic fibrosis and chronic obstructive pulmonary disease patients [17,18]. Its impact on human health is mainly due to the virulence potential it possesses, as well as its intrinsic low susceptibility to a wide assortment of antimicrobial agents and ability to achieve higher resistance levels by mutation [19]. One of the drugs of choice against infections caused by *P. aeruginosa* is fosfomycin, an antibiotic that inhibits cell wall synthesis by targeting the enzyme MurA (UDP-N-acetylglucosamine enolpyruvyl transferase), which catalyzes the first step of peptidoglycan biosynthesis [20]. Concerning the resistance mechanisms that *P. aeruginosa* wields against this drug, the loss-of-function of GlpT, a fosfomycin transporter, may be the most relevant [21]. Additionally, changes in the expression of *fosA*, which codes for an intrinsic fosfomycin-inactivating enzyme [22], together with a heightened activity of an alternative peptidoglycan-recycling pathway that can bypass MurA functions [23,24,25], have been described to contribute to *P. aeruginosa*’s intrinsic resistance to fosfomycin. Conversely, the deletion of genes belonging to said alternative pathway has been suggested to render increased susceptibility to this antibiotic [26].

In former studies, we have determined that *P. aeruginosa* PA14 populations acquire collateral sensitivity to fosfomycin when subjected to adaptive laboratory evolution (ALE) in the presence of various antibiotics [27,28], tobramycin being one of the most appealing, given its prominent role in antipseudomonal therapies [29]. Further, we corroborated the robustness of this trade-off in diverse tobramycin-resistant clones of this strain, although its emergence was contingent on the growth conditions [30,31]. Our results indicated that collateral sensitivity to fosfomycin was caused by a lower expression level of *P. aeruginosa*’s intrinsic mechanisms of fosfomycin resistance. To be precise, *fosA* and the genes encoding the enzymes involved in the alternative peptidoglycan-recycling pathway were expressed at a lower level in tobramycin-resistant/fosfomycin-susceptible mutants than in PA14 strain. As a consequence of reduced FosA activity, the intracellular fosfomycin accumulation of these mutants was higher than that of the wild-type strain, providing a mechanistic explanation of the fosfomycin hypersusceptibility phenotype in these *P. aeruginosa* tobramycin-resistant mutants [31].

With the aim of taking another step forward towards the translation of these results into clinical practice, the study of the conservation of this evolutionary trade-off in *P. aeruginosa* clinical isolates presenting different genomic backgrounds and ab initio resistomes is needed [32]. To that goal, in the current article we have resorted to short-ALE approaches to shed light on the potential collateral sensitivity to fosfomycin of a set of *P. aeruginosa* clinical isolates when acquiring resistance to tobramycin, as well as the mechanisms behind this trade-off. The set encompasses strains belonging to different clonal lineages, including top-ten high risk clones with ubiquitous distribution as sequence type (ST) 11 and ST244; or associated with MDR phenotypes, such as ST274 [33,34]. Our analysis unveiled that collateral sensitivity to fosfomycin after evolution in the presence of tobramycin is contingent on the genomic background: ca. nine of the strains developed this phenotype, whereas the other 14 displayed no changes in their susceptibility to fosfomycin. 

## 2. Results and Discussion

### 2.1. Collateral Sensitivity to Fosfomycin of Pseudomonas aeruginosa Clinical Isolates Evolved in the Presence of Tobramycin Is Not Robust

Previous studies in the laboratory have shown that long-term ALE with tobramycin renders collateral sensitivity to fosfomycin in *P. aeruginosa* PA14 strain [31]. In order to analyze the applicability of this trade-off in clinics, short-term ALE experiments of three days were performed in 23 clinical isolates of this pathogen displaying different genomic backgrounds and ab initio resistomes (Table 1). To note here that short ALE assays with fixed antibiotic concentrations have demonstrated to be more useful than long ones (in which the antibiotic concentration increases over time) when studying the applicability of CS phenotypes [32,35], because they resemble more the conditions of the treated patients and because potential hysteresis situations are avoided [36].

To that goal, we first determined the concentration of tobramycin, close to MIC, that affected—but permitted—the growth of each *P. aeruginosa* clinical isolate in glass tube, to establish the conditions to perform the ALE assays. This information is reported in Appendix A.

Next, the ALE assays were performed for each clinical isolate for three days as stated in the Methods section. Four replicates containing tobramycin, and four controls without antibiotic were included for each isolate, making a total of 184 independent cultures. Once the ALE assays were performed, the susceptibility to tobramycin and fosfomycin of each of the evolved populations, as well as the ones of their corresponding parental strains, was elucidated. The MIC values are reported in Figure 1 and Appendix A. Excepting two populations derived from CLE03-004, tobramycin MICs increased in all cases, up to 10-fold. These data support that tobramycin-resistant populations have been selected after ALE in the presence of tobramycin.

The strains that developed collateral sensitivity to fosfomycin were determined as those that, in at least two out of four of the evolved replicates, showed a fosfomycin’s MIC equal or lower than half of the mean value of the controls’ ones. Then, the results showed that nine out of 23 clinical isolates developed this phenotype of collateral sensitivity to fosfomycin after evolving in the presence of tobramycin. For example, evolved replicate E3 from the clinical isolate AND04-004A showed a MIC of fosfomycin ca. 5-fold lower than the controls’ mean value, and replicate E4 presented a reduction in the MIC of 4-fold. Another relevant reduction in fosfomycin’s MIC was observed in the evolved replicate populations from the clinical isolate CVA01-006, in which three out of the four evolved populations (E2, E3 and E4) showed a MIC of fosfomycin of around an 8-fold reduction regarding the control (Appendix A). Interestingly, in this particular clinical isolate, the MIC to fosfomycin considerably increased in the ALE controls evolved in the absence of tobramycin (ca. 4-fold), so in this strain, some eco-adaptive mutations that improve the growth in medium without antibiotic led to resistance to fosfomycin. The emergence of antibiotic resistance in the absence of selection is an interesting issue has been already described [37], but understanding the causes of this phenotype in this specific strain goes beyond the purposes of this work.

Conversely, 14 out of the 23 clinical isolates tested did not develop collateral sensitivity to fosfomycin, including some evolved populations, as those from ICA01-004, that even showed increased resistance to fosfomycin. This can be due to, among other possibilities, the selection of mutations associated with resistance to fosfomycin, such as the ones in *glpT* [21]; or in genes leading to overexpression of the *fosA*—which encodes a fosfomycin-inactivating enzyme [22]—of genes from the peptidoglycan-recycling pathway, such as *agmK, nagZ, mupP and anmK*, or *murA* [26], encoding the enzyme and catalyzing one of the first steps of the peptidoglycan synthesis and also the target of fosfomycin. 

In conclusion, these results showed that 9 out of 23 clinical isolates, after three days of ALE with tobramycin, developed collateral sensitivity to fosfomycin. So, broadly speaking, as suggested in previous works [38,39], the combination or sequential alternation of tobramycin and fosfomycin to treat *P. aeruginosa* might be worth considering, but, in this case, collateral sensitivity cannot be considered entirely robust and reproducible, because less than half of the tested clinical isolates showed said phenotype. Therefore, the extrapolation of this evolutionary trade-off to treat *P. aeruginosa* infections may be compromised; a valuable piece of information that validates the importance of using clinical isolates when studying these potential therapeutic strategies. However, investigating the isolates that developed collateral sensitivity to fosfomycin, thus delving into the different mechanisms that drove it, was of interest and subsequently addressed.

### 2.2. Principal Component Analysis: The Fold Change in Tobramycin MIC of the Clinical Strains Correlates with the Development of Fosfomycin Collateral Sensitivity

In order to determine if some strain-specific characteristics might be behind the collateral sensitivity to fosfomycin displayed by half of the isolates after the ALE, we performed a PCA [40]. In this case, the PCA was used to divide in a display the clinical isolates into two groups: the ones that showed collateral sensitivity to fosfomycin—according to our criteria—after ALE in the presence of tobramycin and the ones that did not. Therefore, the variables that contributed more to this division would be the ones that most likely had an influence on the development of collateral sensitivity. The ‘best’ two-dimensional representation of a dataset was given by a plot; and the principal components provided a ‘best’ low-dimensional graphical display of the data [41]. In the biplot, presented in Figure 2, the 23 clinical isolates were distributed in a display, following two-dimensional representation, and the variables were plotted too. The variables chosen for the PCA are reported in the Materials and Methods. In the biplot (Figure 2b), the strains were divided into two groups: the ones that developed collateral sensitivity were mostly placed in the bottom part of the display (orange points) and the ones that did not change their fosfomycin susceptibility were in the upper part (blue points). Nevertheless, there was one clinical isolate, ARA02-005, whose evolved replicates showed collateral sensitivity to fosfomycin, but it was placed in the upper part. This is probably due to the fact that the fold change in tobramycin MIC in this strain was lower than in the other isolates (see below).

The analysis of the variables that contributed the most to this distribution showed that the most important was the fold change in the MIC to tobramycin (calculated as the average of MIC of the four evolved replicates in the presence of tobramycin divided by the average of MIC of the four controls evolved in the absence of tobramycin) of the clinical strains, with around 80% of contribution (Appendix A). So, the increase in tobramycin resistance appears to be an important factor in the development of collateral sensitivity to fosfomycin in *P. aeruginosa* clinical strains. In fact, all the evolved populations from the clinical isolates that are present in the bottom-left part of the display presented a high fold change in tobramycin MIC. In addition, the origin of the clinical isolates also had a contribution, around 20%. This variable informs on the source of sampling of the clinical isolates (blood, sputum, or tracheal aspirate; see Appendix A) and could be related, at least to a certain point, to the development of collateral sensitivity to fosfomycin.

Conversely, the rest of the parameters, even the ones that might have been postulated as the most determining a priori, i.e., ST, did not correlate with the collateral sensitivity to fosfomycin the clinical strains acquired, as shown in Appendix A. However, more strains of coinciding STs should be tested in order to fully determine if this parameter can be related to this evolutionary trade-off. In addition, none of the resistome-related mutations appeared to have an influence in the development of collateral sensitivity to fosfomycin after tobramycin ALE, not even the mutations in genes related to resistance to tobramycin, as *fusA1*, which encodes an elongation factor associated with aminoglycosides resistance [42,43]. Indeed, mutations in this gene were reported in the original resistome of FQSE24-0304, which developed collateral sensitivity to fosfomycin, and in GAL02-004 and CLM01-003, which did not. Thence, the original presence of a mutation in *fusA1* did not have an impact on the selection of this trade-off, despite the fold change in the MIC to tobramycin—which could be due to mutations in this gene in the strains with an original non-mutated *fusA1*—contributed more than 80% to that, an interesting dichotomy that would require further research.

### 2.3. The Accumulation of Active Fosfomycin Is Increased in Some Tobramycin-Resistant Mutants Presenting Fosfomycin Collateral Sensitivity and Remains the Same in Mutants without Collateral Sensitivity

In order to unravel whether an increased active fosfomycin accumulation was responsible for fosfomycin collateral sensitivity observed in some clinical isolates evolved in the presence of tobramycin, we estimated the amount of active fosfomycin accumulated inside the cells after incubation with the antibiotic of some of those evolved populations, as described in the Materials and Methods section. We chose the four replicate populations evolved from AND04-004A, the four replicates from FQSE24-0304 and the four replicates of FQSE10-0503 as representatives of those presenting fosfomycin collateral sensitivity, and the four replicates from CAT09-004 and the four from ICA01-004 as representatives of those populations in which fosfomycin collateral sensitivity was not observed. 

The accumulation of active fosfomycin was significantly increased in most replicate populations from FQSE24-0304 and in one replicate population of AND04-004A, while none of the evolved replicates from FQSE10-0503, CAT09-004 and ICA01-004 accumulated a significant higher amount of active fosfomycin when incubated with this antibiotic as compared with their respective controls (Figure 3). These results indicate that an increased active fosfomycin accumulation may be responsible for the fosfomycin collateral sensitivity of some of the tested populations, while different mechanisms might be also contributing to this phenotype, whose study was addressed by measuring the expression of genes encoding intrinsic fosfomycin resistance determinants (see below).

### 2.4. Reduced Expression of Genes Involved in the Alternative Peptidoglycan-Recycling Pathway May Contribute to Fosfomycin Collateral Sensitivity in P. aeruginosa Clinical Isolates

It has been described that increased susceptibility to fosfomycin can be caused, among other mechanisms, by a reduced expression of (i) *fosA*, a gene coding for the fosfomycin-inactivating enzyme, which is intrinsic in *P. aeruginosa* [22,23]; (ii) genes from the alternative peptidoglycan-recycling pathway, as *agmK*, *anmK* and *mupP* [26]; and (iii) *murA*, which encodes a UDP-GlcNAc enol-pyruvyl transferase, important for peptidoglycan synthesis and also the target of fosfomycin, which binds irreversibly to its active site [44]. Thus, with the aim of delving into the reasons behind fosfomycin collateral sensitivity observed in several *P. aeruginosa* clinical isolates, the expression of *fosA, agmK, anmK*, *mupP,* and *murA* was measured in individual clones from evolved populations which showed collateral sensitivity to fosfomycin upon evolving in the presence of tobramycin. Specifically, the evolved populations E4 of AND04-004A, E2 of FQSE24-0304, and E3 of FQSE10-0503 (see Appendix A) were chosen for gene expression analysis, in order to understand alternative mechanisms of fosfomycin collateral sensitivity beyond an increased fosfomycin accumulation in the first case, or further mechanisms in combination with the increased accumulation for the other one (see above). The results, encompassed in Figure 4, showed that the clone derived from the AND04-004A presented a reduced expression of genes from the peptidoglycan-recycling pathway, i.e., *agmK*, with a 0.69-fold change expression with respect to the parental strain; and *anmK*, with an expression significantly lower than the parental one (0.57-fold). The clone derived from FQSE24-0304 presented a statistically significant reduction in the expression of *fosA, anmK,* and *murA* in a 0.4-fold change and of *agmK* in a 0.45-fold change, compared to the parental strain. In addition, the reduced expression of *mupP*, though not statistically significant, was reported with a fold change of 0.61 respect to the parental strain. Notably, the clone derived from FQSE10-0503 presented a reduced expression of *mupP*, with a fold change of 0.64 the parental one, while the expression of the other genes did not change.

Hence, the collateral sensitivity to fosfomycin observed in AND04-004A E4 might be due to the decreased expression of genes from the peptidoglycan-recycling pathway (*agmK* and *anmK*). In contrast, the collateral sensitivity observed in FQSE24-0304 E2 may be ascribed to multiple mechanisms: the aforementioned lessened expression of the genes involved in peptidoglycan recycling; the reduced expression of the gene encoding the fosfomycin-inactivating enzyme—*fosA*—which correlates with the increased active fosfomycin accumulation previously observed in this population (Figure 3); and even the lessened expression of the gene corresponding to the target of fosfomycin: *murA*. Regarding FQSE10-0503, only *mupP* showed a reduced expression as compared to the parental strain, and the levels achieved were not significantly lower. It is important to emphasize that in the clones analyzed, especially in the one from FQSE24-0304 evolved population, the potential underlying mechanisms of fosfomycin collateral sensitivity observed from Laborda and collaborators in PA14 strain [31] agreed with ours. In the work just mentioned, the PA14 clones, evolved with tobramycin, showed a reduced expression of *agmK*, *mupP*, *anmK*, and *fosA*. Consequently, the mechanisms that drive collateral sensitivity seem to coincide, to a certain extent, between *P. aeruginosa* model strain PA14 and different clinical isolates, in which a reduction in the expression of *agmK*, *mupP anmK*, and *fosA* was observed. Therefore, we could affirm that the increased resistance to tobramycin in clinical isolates, acquired after a three days experiment of ALE, can render a down-regulation of the expression of genes from the peptidoglycan-recycling pathway and of the genes that encode the fosfomycin-inactivating enzyme and the target of fosfomycin, in the analyzed clinical isolates of *P. aeruginosa*. This down-regulation of gene expression is likely to entail the collateral sensitivity to fosfomycin on some occasions. Nevertheless, although a lower expression of *mupP* alone, without further decreased expression of other elements of the peptidoglycan recycling pathway—as observed in FQSE10-0503—leads to fosfomycin hypersensitivity [45], the different pattern of expression of the peptidoglycan recycling pathway genes displayed by this mutant in comparison with the other ones supports that different pathways can converge into the phenotype of fosfomycin CS. 

In summary, the results encompassed in this work show useful data on the not-so-promising use of a combinatorial or alternated therapy that pairs tobramycin and fosfomycin against *P. aeruginosa* infections, thereby buttressing the relevance of using diverse clinical strains when postulating therapeutic approaches based on evolutionary trade-offs. Moreover, they shed light on the potential mechanisms behind collateral sensitivity to fosfomycin in this opportunistic pathogen.

## 3. Materials and Methods

### 3.1. Bacterial Strains and Oligonucleotides

The strains and oligonucleotides used in this study are described in Table 1 and Table 2, respectively.

### 3.2. Adaptive Laboratory Evolution

Unless otherwise stated, all the 23 *P. aeruginosa* strains were cultured in Mueller Hinton Broth (MHB) (Sigma, Munich, Germany) at 37 °C and 250 rpm. First, an assay to determine the concentration of tobramycin (close to Minimal Inhibitory Concentration, MIC) that hindered—but allowed—the growth of each *P. aeruginosa* clinical isolate was performed in 2 mL of MHB inoculated with a 1:1000 dilution of overnight cultures. The concentrations tested were determined around the value of MIC for tobramycin reported for the clinical isolates by the laboratory of Dr. Antonio Oliver (Microbiology, Hospital Son Espases, Instituto de Investigación Sanitaria Illes Balears IdISBa). These concentrations were used to carry out the ALE assays, as follows: subsequent culture dilutions of 1/250 were performed each day for 3 days by inoculating 8 µL of culture in 2 mL of MHB. Four replicates containing tobramycin at the specific concentration, which ranged from 0.1 µg/mL to 12 µg/mL (Appendix A), and four controls without antibiotic were included, for each of the analyzed clinical isolates, making a total of 184 independent assays. The tobramycin concentration for each isolate (Appendix A) was maintained over the evolution period. Afterwards, all replicates were preserved in 20% glycerol at −80 °C for further studies.

### 3.3. Analysis of Susceptibility to Antibiotics: E-Test

The susceptibility of the different strains to tobramycin (TOB) and fosfomycin (FOF) was determined using Epsilon-test (E-test) strips (MIC Test Strip, Liofilchem^®^, Waltham, MA, USA) in Mueller Hinton Agar (MHA) (Sigma, Munich, Germany) at 37 °C, following supplier’s instructions. The plates were incubated 24 h at 37 °C to read the MIC values. 

### 3.4. Principal Component Analysis

Principal Component Analysis (PCA) was performed with R software [46] 4.2.2, and a log-transformation using the prcomp function from the stats package was made in order to standardize variables. The biplots and variables plot were generated with the FactoMineR [47] and factoextra [48] packages. The PCA was performed on all the 23 clinical isolates, using as variables the origin of the strains, the ST, their resistome-related mutations (Table 1), their capacity of developing collateral sensitivity to fosfomycin, the MIC of the parental strain to tobramycin and fosfomycin, the fold change of these MICs after the ALE, and the tobramycin concentration used to perform said ALE. The MICs’ fold change was calculated as the average of MICs of a strain’s replicates evolved in the presence of tobramycin, divided by the average of MICs of the same strain’s replicates evolved without antibiotic. The MICs for tobramycin and fosfomycin and the tobramycin concentrations used in the ALE were classified as “high” or “low” in order to be introduced in the program as binary code. The medians of the parental MICs were used for this classification. Tanking these values into account, the parental fosfomycin MIC was considered “high” if the value was ≥64 µg/mL and “low” if the value was ≤48 µg/mL. The parental MIC for tobramycin was considered high if ≥2 µg/mL and low if ≤1.5 µg/mL. The tobramycin concentrations used in ALE assays were considered high if ≥2 µg/mL and low if ≤1 µg/mL. The MIC values are reported in Appendix A. 

### 3.5. RNA Extraction and qRT-PCR

The RNA extraction was performed from three cultures of each strain as previously described in [31]. Briefly, 20 mL of Lysogeny Broth (LB) (Lennox, Pronadisa, Austin, TX, USA) were inoculated from overnight cultures to a final OD_600_ of 0.01 and grown until exponential phase (OD_600_ = 0.6). Then, the resulting pellet was used for the RNA extraction, which was performed following the protocol of RNeasy mini Kit (QIAGEN, Hilden, Germany). The residual DNA was removed by two DNase treatments: the first one in column (DNase QIAGEN incubated with the sample for 20 min at room temperature) and the second one with Turbo DNA-free (Ambion, Austin, TX, USA), incubated for 1 h at 37 °C. The absence of DNA was checked by PCR, using primers rplU_fw and rplU_rv (Table 2). The cDNA was obtained from 2.5–5 μg of RNA in a final volume of 20 μL, using the High-Capacity cDNA reverse transcription kit (Applied Biosystems, Waltham, MA, USA). qRT-PCR was performed in an ABI Prism 7300 Real-time PCR system (Applied Biosystems, Waltham, MA, USA), using Power SYBR green PCR master mix (Applied Biosystems, Waltham, MA, USA). Regarding cDNA, 50 ng was used for the reaction. The primers (Table 2) were used at a concentration of 10 μM. Primer3 Input software was utilized to design these primers and their efficiency was analyzed by qRT-PCR using serial dilutions of cDNA. All mRNA expression values were determined as the average of three independent biological replicates, each one with three technical replicates. Primers rplU_fw and rplU_rv were used to quantify the expression of the housekeeping gene *rplU* that was used to normalize the results, and the relative expression of each gene was calculated with the 2^−ΔΔCt^ method [49].

### 3.6. Measurement of Intracellular Fosfomycin Accumulation

The determination of fosfomycin accumulation was performed as previously described [31]. Briefly, evolved bacterial strains were grown until reaching OD_600nm_ = 0.6 in 20 mL of LB, then the cultures were subsequently centrifuged and suspended in 1 mL of LB containing 2 mg/mL of fosfomycin. After incubation of bacteria with the antibiotic for 60 min at 37 °C, cells were washed three times with 1 mL of 10 mM Tris pH 7.3, 0.5 mM MgCl_2_ and 150 mM NaCl. The pellet was suspended in 100 µL of 0.85% NaCl, from which 10 µL were serially diluted and plated on LB agar (LBA, 1.5% agar) to estimate the number of colony-forming units (CFUs) after the incubation with the antibiotic. The intracellular content of the cells from the remaining volume was obtained by incubation at 100 °C during 5 min and centrifugation at 13,200 rpm for 10 min. A total of 30 µL of intracellular content were added to paper disks (9 mm, Macherey-Nagel, Düren, Germany), which were placed on LBA plates previously seeded with *Escherichia coli* OmniMAX^TM^ (Invitrogen, Waltham, MA, USA). The halo diameters produced by the intracellular content-soaked disks were measured and the amount of fosfomycin accumulated inside each strain was extrapolated from a standard curve calculated with the halo diameters produced by disks containing known concentrations of the antibiotic. Finally, the amount of accumulated fosfomycin was normalized with the number of CFUs of each strain after being incubated with the antibiotic and is represented as fold change of evolved population in the presence of tobramycin respect to the corresponding control—evolved in the absence of antibiotic—population. 

## 4. Conclusions

The analysis of the robustness of bacterial collateral sensitivity is a needed step to transfer this information to clinical practice. Unfortunately, this robustness has been observed just in a few cases. In the present work, we determine that, although previous work showed that collateral sensitivity to fosfomycin in *P. aeruginosa* is robust in different tobramycin-resistant mutants derived from the same strain, this robustness cannot be extrapolated when it comes to clinical isolates. Less than a half of the clinical strains analyzed here presented fosfomycin collateral sensitivity after evolving in the presence of tobramycin. Notably, PCA showed that the main factor contributing to fosfomycin collateral sensitivity was the fold change in the tobramycin MIC. If bacteria become tobramycin-resistant after treatment with this drug, this criterium could be used to prioritize the use of fosfomycin as a second line of treatment. Further, the identification of more specific markers for these particular strains may be relevant to establish when fosfomycin/tobramycin combinations could be useful for treating *P. aeruginosa* infections, a feature that remains to be established and goes beyond the purposes of the work.

## Figures and Tables

**Figure 1 ijms-24-06892-f001:**
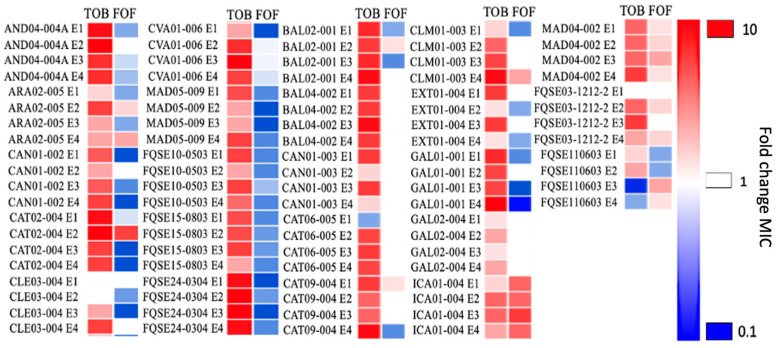
Diagram showing the susceptibility to tobramycin and fosfomycin in clinical isolates of *P. aeruginosa* subjected to short-term ALE in the presence of tobramycin. Susceptibility to tobramycin and fosfomycin was analyzed in 23 clinical isolates (reported in Table 1), 4 replicate populations each—indicated as E1, E2, E3, or E4—subjected to ALE in the presence of tobramycin for three days. Intensity of the color is proportional to fold change of MIC regarding the mean value of the control populations’ MIC. The evolved populations were classified as “resistant” (red) or “susceptible” (blue). Collateral sensitivity to fosfomycin was observed in nine out of the 23 clinical isolates of *P. aeruginosa*. MIC values (μg/mL) of parental strains and populations evolved in the presence/absence of tobramycin, are included in Appendix A. TOB: tobramycin, FOF: fosfomycin.

**Figure 2 ijms-24-06892-f002:**
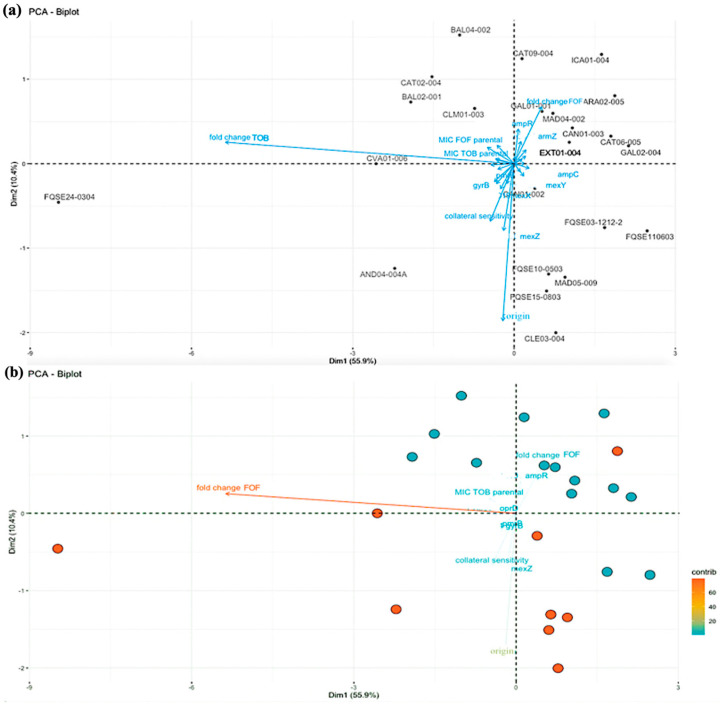
Biplot of Principal Component Analysis of the variables that may influence fosfomycin collateral sensitivity in *P. aeruginosa* clinical isolates. (**a**) In the biplot above the names of clinical isolates and all the variables analyzed are reported. (**b**) In the biplot below the orange points are the strains that developed collateral sensitivity to fosfomycin, while the blue ones are the strains that did not, both after ALE in the presence of tobramycin. In this biplot, only the 10 most influent variables are showed, from the most influential in orange, to the least influential, in blue. The PCA permits to express the information as a set of new variables called principal components; this results in a two-dimension representation (Dim1 and Dim2) of the contribution of all the variables in the division of the strains in two groups. All the variables analyzed by PCA are described in Materials and Methods.

**Figure 3 ijms-24-06892-f003:**
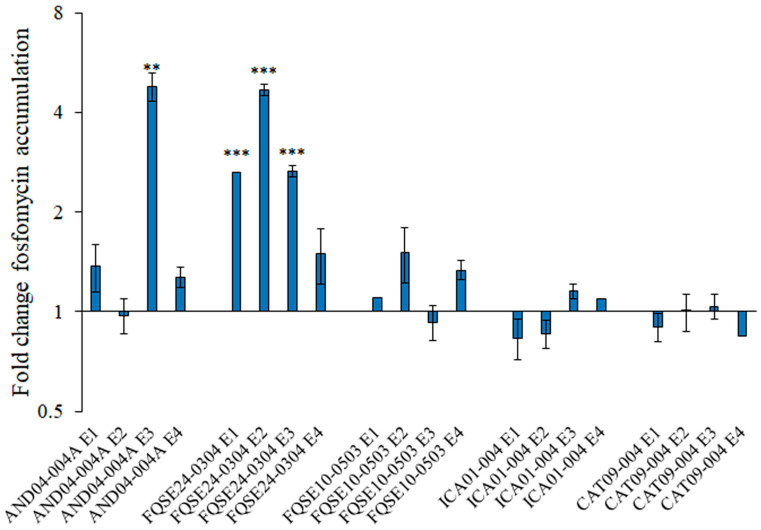
Intracellular active fosfomycin accumulation in populations evolved in the presence of tobramycin from diverse *P. aeruginosa* clinical strains. Fold change of the amount of accumulated active fosfomycin per CFU after incubation of bacteria with 2 mg/mL of the antibiotic in replicate populations evolved in the presence of tobramycin from AND04-004A, FQSE24-0304, FQSE10-0503, ICA01-004 and CAT09-004 strains respect to their respective control—evolved in the absence of antibiotic—populations is represented. The amount of active intracellular fosfomycin is estimated regarding the hallo produced by a disk soaked with the intracellular content of the tested population in a plate seeded with *E. coli* OmniMAX^TM^. Error bars indicate standard deviations of the results from three biological replicates. Statistically significant differences were calculated with *t*-test for paired samples assuming equal variances: ** *p* < 0.005, *** *p* < 0.0005.

**Figure 4 ijms-24-06892-f004:**
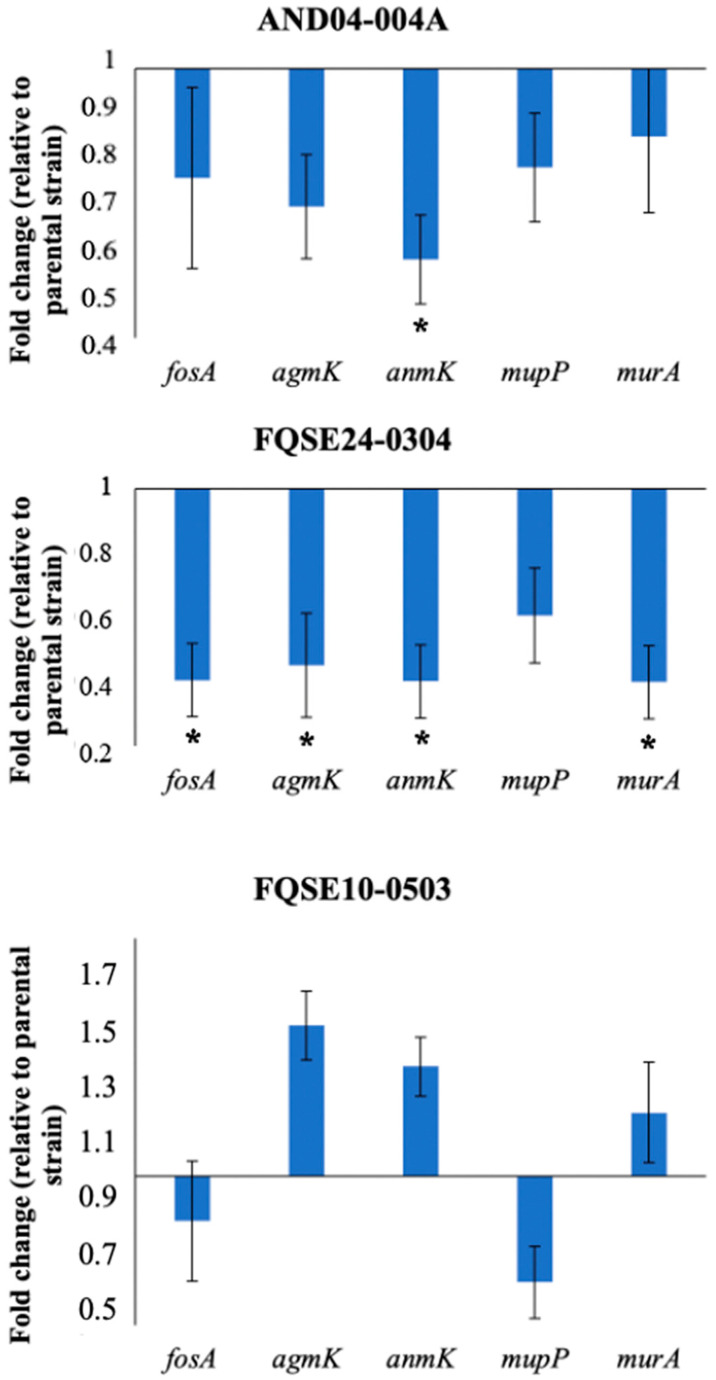
*fosA, agmK, anmK*, *mupP,* and *murA* expression in evolved AND04-004A, FQSE24-0304, and FQSE10-0503 single clones compared with parental ones. Fold change of various genes’ expression level in clones from populations evolved in the presence of tobramycin respect to their parental strains’ is represented. Error bars indicate standard deviations of the results from three biological replicates. Statistically significant differences between the parental strain and the evolved clones were calculated with t-test for paired samples assuming equal variances: * *p* < 0.05.

**Table 1 ijms-24-06892-t001:** Bacterial strains used in this work.

*P. aeruginosa* Clinical Isolates			
Isolate ID	Origin	ST	Mutations in Genes/Proteins Associated with Resistance
AND04-004A	Tracheal aspirate	244	*mexT* (nt234∆8).
ARA02-005	Blood	267	*mexT* (nt234∆8); OprN (R151L); ArmZ (R320C).
BAL02-001	Sputum	1619	*mexT* (nt234∆8).
BAL04-002	Blood	1816	MexA (K86E); *mexT* (nt234∆8); AmpR (G295R); AmpC (A278G); ParE (E215Q).
CAN01-002	Sputum	111	MexB (Q319X); MexY (G530S); MexZ (Q140K); MexT (G276D, nt234∆8);
CAN01-003	Sputum	111	ParS (L137P, nt998∆1); MexY (G530S); *mexT* (nt234∆8); GyrA (T83I, V671I, G860S); Mpl (aa436InsGGF); ParC (S87L).
CAT02-004	Blood	244	*mexR* (nt266InsG); *mexT* (nt234∆8); DacB (PBP4) (A340G).
CAT06-005	Sputum	175	OprM (T198P); OprD (Q142X); MexZ (G195D); *mexT* (nt234∆8); GyrA (T83I, D87N); AmpR (G154R); ParC (S87W, L168Q); ArmZ (V266M).
CAT09-004	Blood	244	*mexT* (nt234∆8).
CLE03-004	Tracheal aspirate	381	*mexZ* (nt386∆1); *mexT* (nt234∆8); AmpC (S173N).
CLM01-003	Sputum	709	GyrB (S466Y); *oprM* (nt501∆5); *mexT* (nt234∆8); AmpR (D135N); FusA1 (Q678L).
CVA01-006	Sputum	175	GyrB (aadB); *mexB* (nt881InsGG); *oprD* (nt55InsGCACT); MexZ (G195D); *mexT* (nt234∆8); GyrA (T83I, D87N); ParC (S87W, L168Q); ArmZ (V266M).
EXT01-004	Sputum	3353	GyrB (S466Y); MexY (D724V); MexS (V308I); *mexT* (nt234∆8); Mpl (P274L); AmpC (D280N, Q311L); FtsI (PBP3) (L219V); AmpD (D83G); OprJ (N121S); MexD (H953N).
GAL01-001	Sputum	560	*mexT* (nt234∆8); GyrA (T83I); *mpl* (nt111InsC).
GAL02-004	Sputum	217	GyrB (R22C); *mexB* (nt3018∆5); MexS (P225L); *mexT* (nt234∆8); GyrA (A908T); Mpl (H190Y); AmpC (V239A); FusA1 (T671A).
ICA01-004	Sputum	698	*mexT* (nt234∆8); MexE (V156A); AmpR (E162Q); ArmZ (V266M).
MAD04-002	Sputum	242	*mexT* (nt234∆8); ParC (K726R)
MAD05-009	Sputum (CF)	27	GyrB (S466F); MexZ (A88P); *mexT* (nt234∆8); AmpC (P274L).
FQSE03-1212-2	Sputum (CF)	274	MexA (L338P); MexZ (A144V); *mexT* (nt234∆8); GyrA (D87N).
FQSE10-0503	Sputum (CF)	274	MexY (V875M); MexZ (IS); *mexT* (nt234∆8).
FQSE110603	Sputum (CF)	701	MexB (473962delC); MexY (N709H,A586T); MexX (A38P); *mexT* (nt234∆8); OprN (R363H); AmpDh2 (P116S).
FQSE15-0803	Sputum (CF)	274	GyrB (E788A); MexA (L338P); MexX (D346H); MexZ (A144V); *mexT* (nt234∆8); PmrB (E213D);
FQSE24-0304	Sputum (CF)	1089	GyrB (S466F); MexA (L338P); OprM (E456G); OprD (V67*); MexY (Y355H); MexX (D346H); MexZ (A194P); GalU (P123L); *mexT* (nt234∆8); *mexF* (Nt_1051_ins9); FusA1 (K430E); PmrB (R287Q).

ST: sequence type, CF: cystic fibrosis.

**Table 2 ijms-24-06892-t002:** Oligonucleotides used in this work.

Oligonucleotides	
Name	Sequence 5′-3′
murA_fw	CATTTCCGGCGCGAAGAACT
murA_rv	ATGCTGCTGGCGTCGACTTCGA
agmK_fw	AGCTGAATCGCTGGTTGGAC
agmK_rv	AACGGTCGGCAGTCTTCCTG
nagZ_fw	AGGTGGGCGGGCTGATCATCTT
nagZ_rv	ATTGGGGTTGTCGGCGATCG
mupP_fw	GCCGGACTTCATCGCCATCA
mupP_rv	AATGCTCCTGGTAGCGGTCGAG
fosA_fw	ACCAGGGCGCCTATCTCGAA
fosA_rv	CGCTGCGGTTCTGCTTCCAT
anmK_fw	CAACGTGCTGATGGACGCCT
anmK_rv	AGCCAGGACAGGTTGAAGCG
rplU_fw	CGCAGTGATTGTTACCGGTG
rplU_rv	AGGCCTGAATGCCGGTGATC

## Data Availability

All data used in the work have been included in the manuscript.

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
