# Peer review of "Collateral Sensitivity to Fosfomycin of Tobramycin-Resistant Mutants of Pseudomonas aeruginosa Is Contingent on Bacterial Genomic Background"

_ijms, 2023, doi:10.3390/ijms24086892_

Round 1
Reviewer 1 Report
The paper by Genova and collaborators reports on the analysis of the robustness of bacterial collateral sensitivity to fosfomycin in tobramycin resistant P. aeruginosa strains by performing a short-ALE experiment on clinical isolates with different ab initio resistomes and different origins.
This work is of high interest because this kind of information can be transferred on clinical practice and in principle can drive treatment of tobramycin resistant infections.
The main conclusion of this work is that that the most relevant parameter influencing collateral sensitivity to fosfomycin is the fold change in the MIC to tobramycin but that, differently to what suggested by a previous work performed on the reference P. aeruginosa PA14 strain, the collateral sensitivity to fosfomycin is not robust in the case tobramycin resistant clinical isolates.
The work is well-written and well-conceived, I have only two criticisms:
-the first is about the duration of the evolution experiment: is it acceptable to compare a short-term experiment with the long-term previous one? Do we expect any difference in the results?
The second one concerns the small number of fosfomycin sensitive isolates used in the gene expression experiment shown in figure 4 (considering the low statistical significance of the differences observed for strain AND04-004A) that prevents to formulate a clear conclusion regarding the mechanism of increased collateral sensitivity. Wouldn’t be possible to expand the analysis to another strain?
Minor point
Line 130: the authors state that collateral sensitivity was observed in 10 out of 23 isolates but at lines 100 and 135 they state that it was shown in 9 isolates. Please correct or explain.
Author Response
The work is well-written and well-conceived, I have only two criticisms:
Answer: we appreciate the positive opinion of the referee regarding our work
-the first is about the duration of the evolution experiment: is it acceptable to compare a short-term experiment with the long-term previous one? Do we expect any difference in the results?
Answer: This is a good point. Long experiments, increasing the concentration of antibiotic in each step allow a better detection of multiple combinations of antibiotic resistance mutations. However, we and others have found that, for testing the emergence of robust CS patterns, shorter evolutions at fixed antibiotic concentrations are better because CS can be decoupled from potential hysteresis and because these conditions resemble more the situation of the treated patient. A short paragraph stating this situation and the required references have been added.
The second one concerns the small number of fosfomycin sensitive isolates used in the gene expression experiment shown in figure 4 (considering the low statistical significance of the differences observed for strain AND04-004A) that prevents to formulate a clear conclusion regarding the mechanism of increased collateral sensitivity. Wouldn’t be possible to expand the analysis to another strain?
Answer: These strains were chosen just as examples to address if the same mechanisms previously described were also found in clinical isolates. The intention was not making an extensive study of all potential mechanisms. Nevertheless, following referees' statements, we have included another strain, not only in the expression studies, but also in the studies of accumulation. Nicely, the accumulation of the antibiotic does not change, a feature that correlates with the absences of changes in the level of expression of fosA, and the only gene showing changes in the level of expression is murP. These findings have allowed to enrich the discussion stating that CS to fosfomycin can be acquired thorough different routes. We thank the referee for the query, because, as stated, the discussion, and hence the article is stronger.
Minor point
Line 130: the authors state that collateral sensitivity was observed in 10 out of 23 isolates but at lines 100 and 135 they state that it was shown in 9 isolates. Please correct or explain.
Answer: We wish like thanking the referee for the observation, because this is a mistake. The right number is 9 and the text has been corrected accordingly.
Reviewer 2 Report
Genova and colleagues found that fosfomycin sensitivity increased in tobramycin-resistant mutants of clinical P. aeruginosa isolates. The manuscript was well written but needs minor corrections of languages in the title and text. For example, Pseudomonas aeruginosa tobramycin-resistant mutants can be changed to tobramycin-resistant mutants of Pseudomonas aeruginosa. They showed that the collateral sensitivity of tobramycin-resistant mutants changed phenotypes with low expression of fosA (intracellular accumulation) and peptidoglycan-recycling enzymes. There might be a tradeoff in the antimicrobial susceptibility of tobramycin-resistant mutants with fosfomycin sensitivity. Authors need to analyze the growth rates of wild-type strains and tobramycin-resistant mutants in the presence or absence of sublethal-dose tobramycin, showing a deleterious or beneficial effect for the selection of P. aeruginosa populations with identified mutations (Figure 1).

Author Response
Genova and colleagues found that fosfomycin sensitivity increased in tobramycin-resistant mutants of clinical P. aeruginosa isolates. The manuscript was well written but needs minor corrections of languages in the title and text. For example, Pseudomonas aeruginosa tobramycin-resistant mutants can be changed to tobramycin-resistant mutants of Pseudomonas aeruginosa. They showed that the collateral sensitivity of tobramycin-resistant mutants changed phenotypes with low expression of fosA (intracellular accumulation) and peptidoglycan-recycling enzymes. There might be a tradeoff in the antimicrobial susceptibility of tobramycin-resistant mutants with fosfomycin sensitivity. Authors need to analyze the growth rates of wild-type strains and tobramycin-resistant mutants in the presence or absence of sublethal-dose tobramycin, showing a deleterious or beneficial effect for the selection of P. aeruginosa populations with identified mutations (Figure 1).
Answer: We appreciate the suggestion of the referee and have changed the title accordingly. We have also revised English along the manuscript following referee's statements. Regarding the need of analyzing growth rates to address if the mutations are benefitial or deleterious concerning tobramysin, what we performed were MIC assays, which is the golden standard methodology to asses if one mutation is benefitial or deleterious to the presence of one antibiotic, We found that, excepting two populations derived from CLE03-004, tobramycin MICs increased in all cases, up to 10 fold in ocassions. This information is shown in Table S2. Growth curves in presence of the antibiotics are needed when no clear changes in MICs are found, but otherwise are not needed because differences in MICs is the regular way of measuring differences " deleterious or beneficial effect for the selection" of antibiotic reistance mutants. Nevertheless, it is true that we included the Figure but did not discuss what the Figure shows. This is misleading and may make the message confuse, not just for the referee, but for any other reader. Consequently, we have introduced a new paragraph discussing the results of the Figure. We appreciate the comment of the referee that served us to clarify this aspect of the article.
Round 2
Reviewer 1 Report
I acknowledge the efforts of the authors to meet my concerns and I'm satisfied by their answers.
In this form the paper can be accepted.
Reviewer 2 Report
Genova et al. revised the manuscript properly according to the reviewer's suggestion.
Only a minor point is, 'commas' in numbers in the y-axis of Figure 4 must be changed to 'dots'.